# Genetic Investigation of Consanguineous Pakistani Families Segregating Rare Spinocerebellar Disorders

**DOI:** 10.3390/genes14071404

**Published:** 2023-07-06

**Authors:** Saadia Maryam Saadi, Elisa Cali, Lubaba Bintee Khalid, Hammad Yousaf, Ghazala Zafar, Haq Nawaz Khan, Muhammad Sher, Barbara Vona, Uzma Abdullah, Naveed Altaf Malik, Joakim Klar, Stephanie Efthymiou, Niklas Dahl, Henry Houlden, Mathias Toft, Shahid Mahmood Baig, Ambrin Fatima, Zafar Iqbal

**Affiliations:** 1Human Molecular Genetics Laboratory, Health Biotechnology Division, National Institute for Biotechnology and Genetic Engineering College (NIBGE-C), Pakistan Institute of Engineering and Applied Sciences (PIEAS), Islamabad 44000, Pakistan; saadia.m.saadi@gmail.com (S.M.S.); hychaudhary@gmail.com (H.Y.); naveednibge@gmail.com (N.A.M.); shahid_baig2002@yahoo.com (S.M.B.); 2Department of Neuromuscular Disorders, UCL Queen Square Institute of Neurology, London WC1N 3BG, UK; e.cali@ucl.ac.uk (E.C.); s.efthymiou@ucl.ac.uk (S.E.); h.houlden@ucl.ac.uk (H.H.); 3Department of Biological and Biomedical Sciences, Aga Khan University, Karachi 74000, Pakistan; lubaba.khalid@scholar.aku.edu (L.B.K.); ghazala.zafar@aku.edu (G.Z.); haqnawaz.khan@aku.edu (H.N.K.); 4Department of Allied Health Sciences, Iqra National University Swat Campus, Swat 19200, Pakistan; sher_buneri@yahoo.com; 5Institute of Human Genetics, University Medical Center Göttingen, 37073 Göttingen, Germany; barbara.vona@med.uni-goettingen.de; 6Institute for Auditory Neuroscience and InnerEarLab, University Medical Center Göttingen, 37075 Göttingen, Germany; 7University Institute of Biochemistry and Biotechnology (UIBB), Pir Mehr Ali Shah Arid Agriculture University Rawalpindi (PMAS-AAUR), Rawalpindi 46300, Pakistan; uzma.abdullah@uaar.edu.pk; 8Department of Immunology, Genetics and Pathology, Uppsala University and Science for Life Laboratory, P.O. Box 815, 751 08 Uppsala, Sweden; joakim.klar@igp.uu.se (J.K.); niklas.dahl@igp.uu.se (N.D.); 9Institute of Clinical Medicine, University of Oslo, P.O. Box 1171, N-0318 Oslo, Norway; mathias.toft@medisin.uio.no; 10Department of Neurology, Oslo University Hospital, P.O. Box 4950 Nydalen, N-0424 Oslo, Norway

**Keywords:** spinocerebellar, consanguinity, ataxia, spastic paraplegia, neurological disorders

## Abstract

Spinocerebellar disorders are a vast group of rare neurogenetic conditions, generally characterized by overlapping clinical symptoms including progressive cerebellar ataxia, spastic paraparesis, cognitive deficiencies, skeletal/muscular and ocular abnormalities. The objective of the present study is to identify the underlying genetic causes of the rare spinocerebellar disorders in the Pakistani population. Herein, nine consanguineous families presenting different spinocerebellar phenotypes have been investigated using whole exome sequencing. Sanger sequencing was performed for segregation analysis in all the available individuals of each family. The molecular analysis of these families identified six novel pathogenic/likely pathogenic variants; *ZFYVE26*: c.1093del, *SACS*: c.1201C>T, *BICD2*: c.2156A>T, *ALS2*: c.2171-3T>G, *ALS2*: c.3145T>A, and *B4GALNT1*: c.334_335dup, and three already reported pathogenic variants; *FA2H*: c.159_176del, *APTX*: c.689T>G, and *SETX*: c.5308_5311del. The clinical features of all patients in each family are concurrent with the already reported cases. Hence, the current study expands the mutation spectrum of rare spinocerebellar disorders and implies the usefulness of next-generation sequencing in combination with clinical investigation for better diagnosis of these overlapping phenotypes.

## 1. Introduction

Spinocerebellar disorders encompass a diverse clinico-genetically heterogeneous continuum of neurodegenerative phenotypes including cerebellar ataxia and hereditary spastic paraplegias. Cerebellar ataxias are a group of movement disorders characterized by progressive degeneration of the cerebellum, causing balance and gait abnormalities [1,2]. Hereditary spastic paraplegias are caused by the degeneration of the corticospinal tract and dorsal column, causing lower-extremity spasticity, hyperreflexia, and extensor plantar responses [3]. Both disease entities may manifest together or with a wide range of remarkable overlapping neurological and non-neurological clinical features that include spastic paraparesis, cognitive impairment, spasticity, dystonia, dysarthria, dysphagia, and ocular abnormalities [4,5]. Globally, the prevalence of spinocerebellar disorders is estimated to be 1 per 10,000 individuals and until now, >200 genetic loci have been associated with these disorders following all modes of monogenic inheritances: autosomal dominant, autosomal recessive, X-linked, and mitochondrial [5,6,7,8,9,10,11].

The classical clinico-genetic strategies for identification of the underlying cause of spinocerebellar disorders are usually constrained due to their highly overlapping nature, however, next-generation sequencing (NGS) technologies provide an unbiased and hypothesis-free approach to determining the underlying disease-causing genetic factors. Since the advent of NGS, the spinocerebellar disorders’ gene discoveries have significantly increased and about 47.3% of these genes have been identified using different NGS platforms [5]. NGS coupled with precise clinical phenotyping is a paradigmatic approach for the identification of genetic variants causing these rare heterogeneous disorders and can help us to understand the underlying patho-mechanisms which can ultimately be used to devise better diagnosis and prognosis strategies as well as enable more specialized and individualized therapies for spinocerebellar disorders.

Herein, whole-exome sequencing was performed to investigate nine unrelated consanguineous Pakistani families presenting with different spinocerebellar phenotypes. The study identified six novel and three previously reported genetic variants in seven genes, thus expanding and strengthening the genotypic-phenotypic spectra of the rare spinocerebellar disorders.

## 2. Materials and Methods

### 2.1. Ethical Approval and Sample Collection

This study was reviewed and approved by the ethics committee of the National Institute for Biotechnology and Genetic Engineering (NIBGE) in Faisalabad, Pakistan, and was conducted in accordance with the principles of the Declaration of Helsinki. Families presenting with rare neurological conditions were recruited from different areas of Pakistan, including Matta Swat, Banu, Faisalabad, Vehari, and Bahawalpur. After written informed consent from parent(s)/guardians, medical history and blood samples were collected from all the available affected and healthy individuals of the families and DNA extraction was performed using standard protocols.

### 2.2. Genetic Analysis

#### 2.2.1. Whole-Exome Sequencing (WES)

To investigate the underlying genetic cause of the disease, WES of the probands from each family was performed. WES of Family A (IV:2, V:1), Family B (V:1), Family G (V:1), Family H (V:1), and Family I (V:1, V:8) was carried out at Novogene Co., Ltd. (Cambridge, UK); whereas WES of Family C (V:1), Family D (V:1), and Family F (IV:5) was performed at Macrogen, Korea using the Agilent SureSelect Human All Exome V6 Kit (Agilent Technologies, Santa Clara, CA, USA) as described elsewhere [12,13]. The Illumina platform NovaSeq 6000 (Illumina, Santa Clara, CA, USA) was used for paired-end (PE150) sequencing. The sequencing reads were aligned against the human reference genome using Burrows-Wheeler Aligner v0.7.17 (BWA). The hg19 reference assembly was used for Family A (IV:2, V:1), Family B (V:1), Family G (V:1), Family H (V:1), and Family I (V:1, V:8); whereas the sequencing reads alignment of Family C (V:1), Family D (V:1), and Family F (IV:5) were performed against hg38 genome assembly. The BAM files generated were sorted and the duplicate reads were marked using SAMtools v1.8 and Picard v2.18.9 (http://sourceforge.net/projects/picard/), respectively. Genotyping was performed with the Genome Analysis Toolkit v4.0 (GATK). Functional annotation and variant filtering were performed with Annotate Variation (ANNOVAR) and FILTUS, respectively. After annotation, the output file was retrieved in the form of a CSV file that was filtered for the identification of potentially pathogenic variants. WES of Family E (V:4, VI:1) was performed using an Ion PI Sequencing 200 Kit (200 bp read length, Life Technologies, Carlsbad, CA, USA) as described elsewhere [14]. The sequencing reads were aligned against hg19 assembly and variant detection was performed using v2.1 of the LifeScope Software (Life Technologies, Carlsbad, CA, USA). Custom R scripts were used to identify potentially damaging variants.

Variant filtration was performed in accordance with the pedigree’s phenotype and pattern of inheritance. Considering that the pathogenic variants are rare in the population, the data variants were filtered against publicly available human polymorphism databases including 1000 Genomes project (https://www.internationalgenome.org/), Genome Aggregation Database (gnomAD, (https://gnomad.broadinstitute.org/), Exome Sequencing Project (ESP, https://evs.gs.washington.edu/EVS/), dbSNP, NHLBI Exome Variant Server, Complete Genomics 69, and priority was given to exonic and splice site variants with a minor allele frequency (MAF) of <1% in these databases. The variants were then prioritized based on their predicted pathogenic impact, i.e., higher weightage was given to frameshift, non-sense, splice site, and missense variants. The deleterious effect of the variants was then assessed using various in-silico prediction tools including Mutation taster, Polyphen-2, SIFT, and CADD scores above 10. Computational assessment of splicing effects used SpliceSiteFinder-like, MaxEntScan, NNSplice, GeneSplicer, ESEfinder, and RESCUE-ESE embedded in Alamut Visual Plus v1.6.1 (Sophia Genetics, Bidart, France) as well as SpliceAI Visual [15]. Further, pathogenic and likely pathogenic variants were prioritized based on the American College of Medical Genetics and Genomics (ACMG) variant classification system. The potential candidate variants were then visually inspected using the Integrative Genomics Viewer (IGV, https://software.broadinstitute.org/software/igv/) to remove any artefacts and false positives.

#### 2.2.2. Homozygosity Mapping

AutoMap (Autozygosity Mapper, Basel, Switzerland) with default parameters was used for the identification of the runs of homozygosity in the whole-exome sequenced individuals except for Family C and Family E [16]. AutoMap offers reliable homozygosity mapping results directly from standard WES outputs (i.e., VCF files). Family C was presenting an autosomal dominant inheritance pattern and homozygosity mapping of Family E was performed manually.

#### 2.2.3. Sanger Sequencing

Sanger sequencing was performed using standard protocols for co-segregation analysis of the candidate pathogenic and likely pathogenic variants. Primers were designed using the Primer blast tool (www.ncbi.nlm.nih.gov/tools/primer-blast/) and the targeted region was amplified. Sequencing analysis was performed using an ABI-3730 DNA analyzer (Applied Biosystems, Waltham, MA, USA) and data obtained were visualized by Sequencher 5.0 software.

## 3. Results

### 3.1. Clinical Findings

In this study, we investigated nine consanguineous Pakistani families presenting distinct neurological features. The clinical findings for all the patients are summarized in Table 1 and their representative images are given in Figure 1.

Family A

Family A is a multiplex family, comprising four affected individuals (IV:2, IV:3, IV:8, V:1) (Figure 2). All the affected individuals presented similar clinical features including progressive abnormal ataxic to no gait, mild to moderate intellectual disability, muscle atrophy, and dysarthria. Patient IV:3 has a more severe phenotype compared to other affected individuals and manifested hip dysplasia.

Family B

Family B has three affected individuals (V:1, V:2, V:3) and five healthy siblings born to consanguineous parents (Figure 2). All the patients presented global developmental delay and ataxic gait.

Family C

This is the only family following an autosomal dominant pattern of inheritance and the loop being investigated comprises five affected individuals including a mother (IV:5), four affected siblings (V:1, V:2, V:3, V:4), and three unaffected siblings (Figure 2). Patients V:1 and V:2 are 17- and 11-year-old females with disease onset at 6 and 4 years of age, respectively. Both patients achieved normal developmental milestones. The clinical features observed in both individuals were gait abnormality, foot deformities including club feet and pes planus, short stature, and hip dysplasia. No signs of cognitive impairment were observed. Patient V:3 is a 12-year-old male, with disease onset at 4 years of age, presenting abnormal gait, foot deformities, hip dysplasia, short stature, and upper limb weakness. Patient V:4 is a six-year-old male, presenting with a more severe phenotype compared to the other affected individuals in the family. The disease onset was at three years of age. He never achieved the walking milestone and presented foot deformities. The mother was also presenting with similar clinical features but with severe hip dysplasia. The course of the disease is non-progressive in all the affected individuals of the family.

Family D

Family D comprises two affected individuals (V:1, V:5) and five healthy siblings born to consanguineous parents (Figure 2). Both affected individuals are males presenting with similar phenotypic features. The symptoms observed were upper and lower limb muscle spasticity, progressive loss of movement, bending of hands and feet, dysarthria, dysphagia, and muscular atrophy. No cognitive deficits were noticed. There is another affected individual (IV:3) in the extended family manifesting similar features, but she died at the age of 25 years due to unknown reasons.

Family E

Family E consisted of six affected individuals (V:4, V:5, V:8, VI:1, VI:2, and VI:3) born to three consanguineous couples (Figure 2). The disease symptoms in all affected individuals started to appear at the age of around one year and are progressive in nature. The patients never attained walking independently and were unable to walk even with support at the age of 4 years. The other symptoms observed were upper motor neuron deterioration, dysarthria, tongue spasticity, dysphagia, hyper-salivation, and bending of hands and feet. No cognition or hearing defects were noticed.

Family F

Family F consisted of five affected individuals (IV:1, IV:4, IV:5, IV:6, IV:7), dispersed in two different loops and born to first cousins (Figure 2). The disease onset is between 6–8 years of age and the course of the disease is progressive. Patient IV:1 manifested ataxic gait and mild cognitive dysfunction. Patients IV:5 and IV:7 presented ataxic gait, cognitive dysfunction, and dysarthria. Patient IV:6 was wheelchair-bound/bedridden, presenting with similar clinical features of cognitive dysfunction and dysarthria.

Family G

Family G comprises three affected individuals, two females (V:1, V:2), one male (V:3), and one healthy sibling born to first cousins (Figure 2). The patients presented with the typical clinical features of the disease including ataxic gait, lower limb muscle atrophy, and brisk reflexes. The course of the disease was progressive and onset was around 3 years of age in all affected individuals. Dysarthria and planter reflex were observed in patients V:2 and V:3, while the proband (V:1) showed no planter reflex and normal speech. Moderate cognition impairment was also observed in all the affected individuals. No seizures were manifested by any of the patients.

Family H

Family H has four affected individuals, three males (V:1, V:2, V:4) and one female (V:3) who were born to consanguineous parents (Figure 2). The disease onset was around 16 years of age. All the patients were manifesting a similar clinical phenotype. Initially, gait disturbance was observed followed by wheelchair dependence. No ocular abnormality was present in the patients. Mild to moderate intellectual disability was observed in all affected individuals. The course of the disease is progressive.

Family I

Family I is a multiplex consanguineous family with six affected individuals (V:1, V:4, V:8, V:11, IV:7, IV:8), dispersed in four different loops (Figure 2). Patient V:1 was a 32-year-old male, manifesting progressive ataxic gait, action tremor, dysarthria, slight dysmetria, and nystagmus. He achieved normal developmental milestones and the disease onset was at 12 years of age. The patient had a history of head trauma with cranial surgery at the age of 17 years and experienced febrile seizures in childhood until the age of 7–8 years. Patient V:4 was a 27-year-old female with disease onset at 12 years, presenting progressive ataxic gait with supported walking, action tremor, dysarthria, and dysmetria. Patient V:8 was a 37-year-old male, who achieved normal developmental milestones. The disease onset was at 15 years. High steppage gait with support, choreic movements, nystagmus, and clawed fingers/toes were observed in the patient. Patient V:11 is a 23-year-old female with disease onset at 16 years of age, presenting with broad-based ataxic gait, choreic movements, weak eyesight, and fainting episodes.

### 3.2. Molecular Findings

Genetic investigation of these families revealed that novel disease-causing variants are segregating in Families A–F; *ZFYVE26*: c.1093del (p.Leu365SerfsTer16), *SACS*: c.1201C>T (p.Arg401Ter), *BICD2*: c.2156A>T (p.Lys719Met), *ALS2*: c.2171-3T>G, *ALS2*: c.3145T>A (p.Tyr1049Asn), and *B4GALNT1*: c.334_335dup; (p.Ala113GlyfsTer27), respectively (Figure 2, Appendix A).

Data analysis of Families G, H, and I revealed previously reported variants *FA2H*: c.159_176del (p.53_58del), *APTX*: c.689T>G (p.Val230Gly), and *SETX*: c.5308_5311del (p.Glu1770fs), respectively [17,18,19,20] (Figure 2, Appendix A). According to ACMG guidelines, all variants are classified as either pathogenic or likely pathogenic except the *ALS2* variants, which are classified as variants of uncertain significance (VUS). The list of filtered variants in all the families is available in Appendix A. Importantly, the c.2171-3T>G variant is predicted to cause cryptic acceptor site activation that would likely result in a frameshift through the inclusion of two nucleotides and a slight shifting of exonic splice enhancer (ESE) hexamers (Figure 3). The in silico prediction tools predicted all of the identified variants to be deleterious (Table 2).

Retrospectively, homozygosity mapping performed manually for Family E (individuals V:4, VI:1) (Appendix A) and using VCF files of all the other affected individuals (except Family C) that are subjected to WES have identified the homozygous regions encompassing the identified rare homozygous variants in this study (Appendix A).

## 4. Discussion

The human brain is a complex entity, the development and functionality of which are regulated by the genetic code. The variation in this genetic code can give rise to various disease conditions including neurological conditions/diseases with potentially lifelong consequences. The global prevalence rate of these life-threatening ailments was estimated to be 10.2%, furthermore, the causality rate was also found to be very high, i.e., approximately 16.8% [21]. These disorders manifest with immense clinical variability, often with overlapping phenotypic features and genetic heterogeneity, hence posing a great challenge to identifying the underlying genetic cause of the disease and understanding the involved patho-mechanisms. Consanguineous unions serve as the most likely framework for the study of genetic disorders with a recessive mode of inheritance, as consanguinity increases the chances of inheriting pathogenic variants in the homoallelic state from parents to offspring. Pakistan is the fifth most populous country and has a very high rate of consanguinity, with more than 60% consanguineous unions [22]. Thus, the Pakistani population presents a unique opportunity to discover the recessive genetic causes of rare inherited disorders. Currently, NGS is the highly pertinent approach for identifying the underlying genetic cause of rare disorders in both research and clinical settings, leading to the resolution of the diagnostic odyssey. However, in Pakistan, availability of the limited resources restrains the use of NGS for diagnostic purposes.

Herein, nine unrelated consanguineous families were studied. The families were manifesting a range of spinocerebellar phenotypes including spastic paraplegia 15 (Family A), spastic ataxia (Family B), spinal muscular atrophy with lower extremity predominance type 2 (Family C), infantile ascending hereditary spastic paraplegia (Family D and E), spastic paraplegia 26 (Family F), spastic paraplegia 35 (Family G), ataxia with oculomotor apraxia type 1 (Family H), and ataxia with oculomotor apraxia type 2/spinocerebellar ataxia with axonal neuropathy 2 (Family I). They were investigated for the identification of the genetic basis of the disease by employing whole-exome sequencing followed by Sanger sequencing.

In our cohort, there are a total of 31 affected individuals, manifesting overlapping neurological features similar to the already reported cases. The key clinical symptoms observed are gait abnormalities or no ambulation in 31/31 patients, cognitive deficit in 11/31 patients, and muscular atrophy in 12/31 patients. The additional clinical features of all patients are summarized in Table 1, and are in line with the established clinical presentation of the respective diseases.

In our study, we identified four novel pathogenic variants in *ZFYVE26, SACS, BICD2,* and *B4GALNT1*, two *ALS2* variants of uncertain significance (VUS) in six unrelated consanguineous families (Family A–F), and three previously reported variants in *FA2H*, *APTX,* and *SETX* in Families G–I [17,18,19,20]. The *ALS2* VUS are considered as the underlying cause of the disease because the phenotypic features observed in our patients are consistent with the *ALS2*-related disease, but further functional analysis is required to prove the possible pathogenicity. Family C is following an autosomal dominant inheritance pattern, all of the other families present an autosomal recessive mode of inheritance. In Family C, three likely pathogenic variants have been identified including *BICD2*, *PTPN11,* and *LAMA5*. *BICD2* and *PTPN11* are known to cause autosomal dominant disorders (OMIM ID 609797 and OMIM ID 176876, respectively), whereas *LAMA5* is involved in an autosomal recessive phenotype (OMIM ID 601033). On the basis of the mode of inheritance and the phenotypic features manifested by the affected individuals in our families, we considered the variant affecting *BICD2* as the plausible disease-causing variant.

We identified frameshift biallelic variants in *ZFYVE26*: c.1093del, *B4GALNT1*: c.334_335dup, and *SETX*: c.5308_5311del in Families A, F, and I, respectively, and a novel non-sense variant in *SACS*: c.1201C>T in Family B, potentially resulting in truncated proteins lacking the crucial functional domains and hence playing a role in disease pathology (Appendix A). *ZFYVE26* encodes a zinc-finger protein spastizin that is highly expressed in the brain and is a component of the adaptor-related protein complex 5 (AP5) that plays a crucial role in autophagic lysosomal reformation. Its dysfunctioning can result in the loss of neuronal cells [23,24]. Our study reports the second identified case of *ZFYVE26* related spinocerebellar degeneration in the Pakistani population [25]. *B4GALNT1* translates the β1, 4-N-acetylgalactosaminyl transferase-1 (GalNAcc-T) enzyme involved in the synthesis of sialic acid-containing complex gangliosides (GM2 and GD2) which are present in the plasma membrane of cells, predominantly in neurons. They are involved in signal transduction, synaptic plasticity, and endocytosis; hence crucial for the nervous system [26]. The alteration in the enzyme results in disruption of the ganglioside metabolic pathway and subsequently leads to neurodegenerative lysosomal storage disorders [27,28,29]. *SETX* encodes for the Senataxin protein, which has DNA/RNA helicase activity and is considered to play a vital role in DNA double-strand repair and RNA splicing mechanisms [30].

Family C is segregating the novel heterozygous missense variant in *BICD2*: c.2156A>T in an evolutionarily conserved region and thus possibly affecting the protein’s structure. The *BICD2* (bicaudal D homolog 2) protein is implicated in axonal transport along microtubules and maintains the functional integrity of lower motor neurons. The protein regulates the trafficking of crucial cellular cargoes such as Golgi, secretary vesicles, and mRNA by interacting with the dynein-dynactin complex and small GTPase RAB6. The mutation in *BICD2* results in impaired axonal transport, leading to motor neurodegeneration and causing the disease [31,32].

In Family D, a novel homozygous splice site variant in *ALS2*: c.2171-3T>G located in the splice acceptor site of intron 10 was identified. The c.2171-3T>G variant is predicted to reside in an AG-exclusion-zone, slightly shift ESE hexamers, and abolish the native splice acceptor site, likely resulting in a frameshift through the inclusion of two nucleotides. Whether or not this results in all resulting transcripts showing the predicted frameshift or leaky splicing that includes evidence of other effects such as exon skipping or partial wild-type splicing remains to be determined. The NetGene2-2.42 server (https://services.healthtech.dtu.dk/services/NetGene2-2.42/) predicted that this variant results in the skipping of the wild-type acceptor splice site of intron 10 and thereby activation of the nearby dormant cryptic splice sites, presumably leading to non-sense mediated decay (NMD) of the mutant transcript or aberrant protein production [33,34]. *ALS2* encodes the alsin rho guanine nucleotide exchange factor protein which is highly expressed in the central nervous system, particularly in the cerebellum, and is involved in the activation of small GTPase RAB5, thereby regulating the endosome as well as mitochondrial trafficking and fusions in the neurons [35]. The homozygous missense variants in *ALS2*: c.3145T>A and *APTX*: c.689T>G are identified in Family E and H, respectively. *APTX* encodes for the aprataxin nuclear protein, which is involved in DNA single-strand break repair and is highly expressed in the cerebellum, cerebellar cortex, spinal cord, basal ganglia, and other nervous system tissues [36,37].

The previously reported homozygous 18bp deletion in *FA2H*: c.159_176del was identified in Family H and resulted in the loss of highly conserved amino acid residues, thus affecting protein function and causing the disease. *FA2H* encodes for the lipid biosynthetic enzyme fatty acid 2-hydroxylase, which is involved in the formation of the brain cell’s myelin sheath which protects the neuronal axons from damage and enhances the nerve conduction rate [38]. Recently, three Pakistani families reported the same variant (*FA2H*: c.159_176del), hence classifying it as a founder effect mutation in the Pakistani population [18].

The present study expands the mutation spectrum of spinocerebellar disorders and further strengthens the usefulness of WES as an efficient and convenient approach for identifying the underlying cause of these rare genetic diseases owing to their highly clinically overlapping presentation, particularly in inbred populations such as Pakistan’s. This will also help to improve the diagnosis and prognosis of rare spinocerebellar disorders, thus leading to the establishment of better genetic counselling and carrier screening opportunities and thereby reducing the disease burden.

## Figures and Tables

**Figure 1 genes-14-01404-f001:**
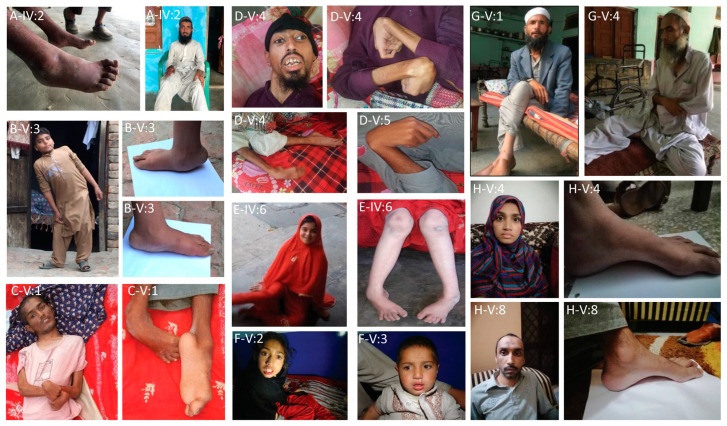
Representative images of selected affected individuals (A-IV:2) Pictures of Family A patients IV:2 (B-V:3) Clinical pictures of Family C. Patients V:3 showing foot deformities; club feet and pes planus (C-V:1) Clinical pictures of Family D. Patient V:1 showing both upper and lower limb spasticity and muscular atrophy (D-V:4–D-V:5) Clinical pictures of Family E. Patients (V:4, V:5) showing limb spasticity and muscular atrophy (E-IV:6) Clinical pictures of Family F. Patient (IV:6) showing lower limb deformities (F-V:2–F-V:3) Clinical pictures of Family G. Patient V:2- and V:3 (G-V:1–G-V:4) Clinical pictures of Family H patients V:1, V:4 (H-V:4–H-V:8) Clinical pictures of Family I, showing foot deformities; pes cavus in patients V:4, V:8.

**Figure 2 genes-14-01404-f002:**
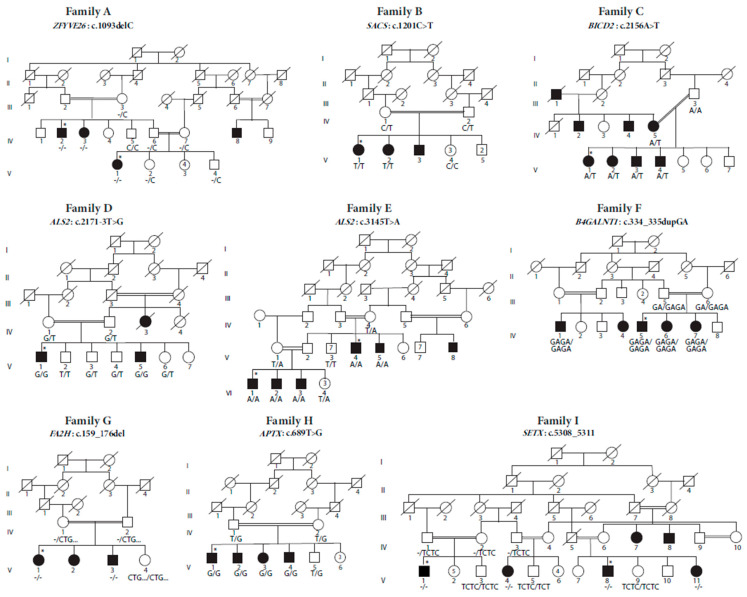
Pedigrees of families A–I, illustrating the segregation of the pathogenic variants in the respective families. Asterisk in each pedigree represents the individual subjected to WES. All the affected individuals are indicated in filled black boxes (males) and circles (females).

**Figure 3 genes-14-01404-f003:**
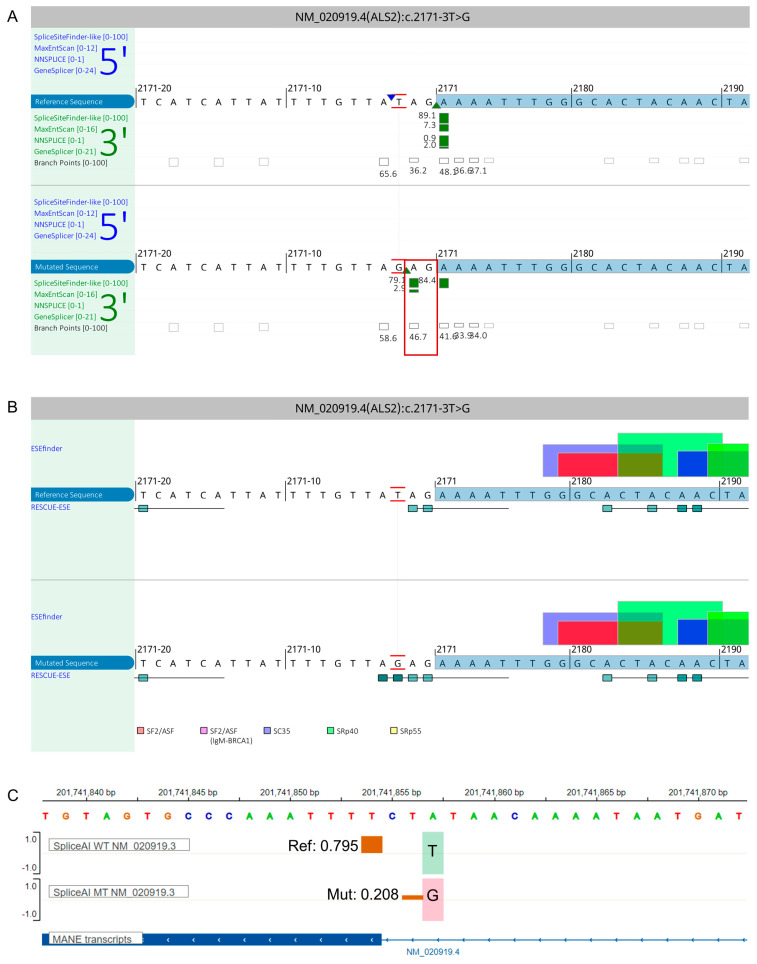
In silico prediction of the ALS2 c.2171-3T>G variant. (**A**) In silico splice prediction of the wild-type (top track, red marked T) and c.2171-3T>G (bottom track, red marked G). Only scores that are predicted to change are shown. The cryptic splice acceptor site that is predicted to be activated and cause the inclusion of AG is boxed in red. (**B**) ESEfinder and RESCUE-ESE analysis of the wild-type and the c.2171-3T>G variant. Analysis with ESEfinder and RESCUE-ESE reveals the splicing sequence landscape for the wild-type (upper panel) and the variant (lower panel) sequence, with the respective c.2171-3 nucleotides marked in red. ESE hits are displayed above each sequence while RESCUE-ESE hexamers are shown below in green boxes and show RESCUE-ESE hexamers. The c. 2171-3T>G variant is predicted to slightly shift and strengthen an ESE hexamer that is indicated by the addition of two dark green boxes in the bottom sub-panel. (**C**) SpliceAI Visual prediction at the splice acceptor region of exon 11 in antisense orientation. Green and pink bars on the right of the exon 11 boundary show the wild-type (T, top track) and mutant base (G, bottom track), respectively. Brown bars show the splice acceptor sites [score range: 0–1]. Here, a score closer to 1 represents a strong splice score. The score must be ≤0.2 in order to reach the threshold. The native splice acceptor site is shown on the left with a score of 0.795. A cryptic splice acceptor site is shown with a smaller brown bar and a score of 0.208. Abbreviations: Ref, reference; Mut, mutant; WT, wild-type.

**Table 1 genes-14-01404-t001:** Clinical findings of affected individuals in Family A–I.

Pedigree	Patient ID	Age (y)	Gender	Onset (y)	Clinical Diagnosis	Symptoms and Signs
Family A	IV:2	28	M	N/A	Spastic paraplegia 15	No ambulation, intellectual disability, muscle atrophy, dysarthria
	IV:3	18	F	N/A		Ataxic gait, intellectual disability, muscle atrophy, dysarthria, hip dysplasia
	V:1	15	F	N/A		Ataxic gait, intellectual disability, muscle atrophy
Family B	V:1	N/A	F	N/A	Spastic Ataxia	Ataxic gait, delayed developmental milestones
	V:2	N/A	F	N/A		Ataxic gait, delayed developmental milestones
	V:3	N/A	M	N/A		Ataxic gait, delayed developmental milestones
Family C	V:1	17	F	6	Spinal muscular atrophy, lower extremity-predominant, 2A, autosomal dominant	Gait abnormality, club feet, pes planus, short stature
	V:2	11	F	4		Gait abnormality, club feet, pes planus, hip dysplasia
	V:3	12	M	4		Gait abnormality, club feet, pes planus, hip dysplasia, upper limbs weakness, short stature, lower motor neuron deterioration symptoms
	V:4	6	M	3		No ambulation, club feet, lower motor neuron deterioration symptoms
Family D	V:1	24	M	1.5	Infantile ascending hereditary spastic paraplegia	Muscle spasticity, Progressive loss of movement, bending of hands and feet, dysarthria, dysphagia, muscular atrophy, no cognitive deficit
	V:5	14	M	5		Muscle spasticity, Progressive loss of movement, bending of hands and feet, dysarthria, dysphagia, muscular atrophy, no cognitive deficit
Family E	V:4	22	M	2	Infantile ascending hereditary spastic paraplegia	Muscle deterioration, progressive loss of movement, bending of hands and feet, dysarthria, dysphagia, hyper salivation, no cognitive deficit
	V:5	25	M	2		Muscle deterioration, progressive loss of movement, bending of hands and feet, dysarthria, dysphagia, hyper salivation, no cognitive deficit
	VI:1	14	M	2		Muscle deterioration, progressive loss of movement, bending of hands and feet, dysarthria, hyper salivation, no cognitive deficit
	VI:2	6	M	2		Muscle deterioration, progressive loss of movement, bending of hands and feet, hyper salivation, no cognitive deficit
	VI:3	3	M	2		Muscle deterioration, progressive loss of movement, hyper salivation, no cognitive deficit
Family F	IV:1	26	M	7	Spastic paraplegia 26	Ataxic gait, mild intellectual disability
	IV:5	20	M	6		Ataxic gait, dysarthria, mild intellectual disability
	IV:6	15	F	6		No ambulation, intellectual disability, dysarthria, in toeing
	IV:7	10	F	8		Ataxic gait, mild intellectual disability
Family G	V:1	10	F	3	Spastic paraplegia 35	No ambulation, hyperreflexia, lower limbs muscle atrophy
	V:2	4	F	3		Ataxic gait, hyperreflexia, lower limbs muscle atrophy
	V:3	3	M	3		Mild ataxic gait
Family H	V:1	40	M	16	Ataxia with oculomotor apraxia type 1	Progressive loss of ambulation, intellectual disability
	V:2	37	M	15		Progressive loss of ambulation, intellectual disability
	V:3	34	F	16		Progressive loss of ambulation, intellectual disability
	V:4	51	M	16		Progressive loss of ambulation, intellectual disability
Family I	V:1	32	M	12	Ataxia with oculomotor apraxia type 2/ Spinocerebellar ataxia with axonal neuropathy 2	Progressive ataxic gait, experienced febrile seizure till 8 years of age, action tremor, dysarthria, nystagmus
	V:8	37	M	15		Progressive ataxic gait, choreic movement, nystagmus
	V:11	23	F	16		Progressive ataxic gait, choreic movements, weak eyesight, fainting episodes

N/A = Not Available.

**Table 2 genes-14-01404-t002:** Variant Annotation, Allele Frequencies, and in silico Predictions of the Variants Identified in Family A–I.

		Family A	Family B	Family C	Family D	Family E	Family F	Family G	Family H	Family I
Variant Annotation	Gene	*ZFYVE26*	*SACS*	*BICD2*	*ALS2*	*ALS2*	*B4GALNT1*	*FA2H*	*APTX*	*SETX*
	Transcript	ENST00000347230.9	ENST00000682944.1	ENST00000356884.10	ENST00000264276.10	ENST00000264276.11	ENST00000341156.8	ENST00000219368.8	ENST00000379817.7	ENST00000224140.6
	GRCh38/hg38 position (DNA change)	14:68272260	13:23355411	9:92717899	2:201741857	2:201726701	12:57631248	16:74774580	9:32984712	9:135187207
	cDNA change	c.1093del	c.1201C>T	c.2156A>T	c.2171-3T>G	c.3145T>A	c.334_335dup	c.159_176del	c.689T>G	c.5308_5311del
	Protein change	p.Leu365SerfsTer16	p.Arg401Ter	p.Lys719Met	N/A	p.Tyr1049Asn	p.Ala113GlyfsTer27	p.53_58del	p.Val230Gly	p.Glu1770fs
	Variant type	Frameshift	Stop	Missense	Splice site	Missense	Frameshift	Inframe 18bp del	Missense	Frameshift
	Zygosity	Homozygous	Homozygous	Heterozygous	Homozygous	Homozygous	Homozygous	Homozygous	Homozygous	Homozygous
Allele Frequencies	dbSNP ID	N/A	rs769212398	N/A	N/A	N/A	N/A	rs759947457	rs536584919	rs750959420
	gnomADv3 (highest subpopulation)	N/A	0.00000658	N/A	N/A	N/A	N/A	N/A	0.0000132	N/A
	gnomADv2.1 (highest subpopulation)	N/A	0.00000398	N/A	N/A	N/A	N/A	N/A	0.0000159	0.0000318
	Ensembl browser	N/A	N/A	N/A	N/A	N/A	N/A	N/A	0.0002	N/A
	Iranome	N/A	N/A	N/A	N/A	N/A	N/A	N/A	N/A	N/A
	GME Variome	N/A	N/A	N/A	N/A	N/A	N/A	N/A	N/A	N/A
	ACMG Classification	Pathogenic (PVS1, PM2, PP1, PP4)	Pathogenic (PVS1, PM2, PP1, PP3,PP5)	Likely Pathogenic (PM1, PM2, PP1, PP2, PP3, PP4)	Uncertain Significance (PP1, PP3, PP4, PM2)	Uncertain Significance (PM2, PP1, PP3, PP4)	Pathogenic (PVS1, PM2, PP1, PP4)	Pathogenic (PS3, PM1, PM4, PM2, PP1, PP4)	Pathogenic (PS1, PM1, PM2, PP1, PP3, PP4)	Pathogenic (PVS1, PM2, PP1, PP4)
Insilico Predictions	GERP++RS	N/A	5.72	N/A	N/A	5.72	N/A	N/A	5.59	N/A
	CADD	N/A	36	31	23.2	29	N/A	N/A	27.4	32
	Polyphen-2	N/A	N/A	0.999	N/A	1	N/A	N/A	1	N/A
	SIFT	N/A	N/A	0	N/A	0	N/A	N/A	0	N/A
	Mutation Taster	1	1	1	1	1	1	1	1	1

N/A = Not Available.

## Data Availability

The data from this study can be made available upon request.

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
