# Peer review of "Genetic Investigation of Consanguineous Pakistani Families Segregating Rare Spinocerebellar Disorders"

_genes, 2023, doi:10.3390/genes14071404_

Round 1

Reviewer 1 Report

Saadi et al. wrote a manuscript entitled "Genetic investigation of cosanguineos Pakistani families segregating rare spinocerebellar dsisorders".  This is a very interesting study with scientifically robust results, expanding the spectrum of mutations of rare forms of spinocerebellar ataxias, with the respective phenotypes, fully demonstrating the importance of the next generation sequencing exams.

Author Response

Thank you very much for acknowledgments

Reviewer 2 Report

The manuscript of Saadi et al. presents genetic causes of the rare spinocerebellar disorders in nine consanguineous Pakistani families. The clinical features of all patients in each family are similar to the already reported cases, therefore my main concern is the novelty of the manuscript. However, the cases are clearly presented and derive from the cosanguineous populations in Pakistan.

The introduction is concise and presents the most important aspects of spinocerebellar disorders, including symptoms and diagnostics with NGS. The results are presented clearly, there are illustrated by figures and supplementary materials. The disussion is unclear for me in one paragraph, I have listed it in the comments. Th conclusions are short and summarize the results.

Comments:

Line 76-please list the areas of Pakistan where the families come from if possible

Figure 1 the photos ar small and therefore not every detail can be seen. I would either make them bigger or focus on the most important aspects.

Discussion: lines 18-27- the concept of genetic blueprint and what exactly is meant is unclear to me. Please explain or reformulate

I would rather put highlight on the consanguinity and diagnostics of such kind of diseases in Pakistan in the disussion. Are there any other studies on the disgnostic yield of NGS in cosanguineous families? What is the standard diagnstics in Pakistan? Are there any statistics on how many families in Pskistan are cosanguineous?

Author Response

Comments and Suggestions for Authors

The manuscript of Saadi et al. presents genetic causes of the rare spinocerebellar disorders in nine consanguineous Pakistani families. The clinical features of all patients in each family are similar to the already reported cases, therefore my main concern is the novelty of the manuscript. However, the cases are clearly presented and derive from the cosanguineous populations in Pakistan.

The introduction is concise and presents the most important aspects of spinocerebellar disorders, including symptoms and diagnostics with NGS. The results are presented clearly, there are illustrated by figures and supplementary materials. The disussion is unclear for me in one paragraph, I have listed it in the comments. Th conclusions are short and summarize the results.

Comments:

  1. Line 76-please list the areas of Pakistan where the families come from if possible

Response: Families presenting rare neurological conditions were recruited from different areas of Pakistan, including Matta Swat, Banu, Faisalabad, Vehari, and Bahawalpur., Please have a look at lines 76 and 77.

  1. Figure 1 the photos are small and therefore not every detail can be seen. I would either make them bigger or focus on the most important aspects.

Response: Thanks for the suggestion, we have modified the figure1

  1. Discussion: lines 18-27- the concept of genetic blueprint and what exactly is meant is unclear to me. Please explain or reformulate

Response: Genetic blue print means the genetic make-up (DNA)/genetic code. However, we have reformulated, and used genetic code instead of genetic blueprint. Please have a look at lines 20 and 21 in the discussion part.

  1. I would rather put highlight on the consanguinity and diagnostics of such kind of diseases in Pakistan in the disussion. Are there any other studies on the disgnostic yield of NGS in cosanguineous families? What is the standard diagnstics in Pakistan? Are there any statistics on how many families in Pskistan are cosanguineous?

Response: Pakistan is the fifth most populous country and have a very high rate of consanguinity, with more than 60% consanguineous unions. Thus, Pakistani population presents a unique opportunity to discover the recessive genetic causes of rare inherited disorders. Currently, NGS is the highly pertinent approach for identifying the underlying genetic cause of the rare disorders in both research and clinical settings, leading to resolve the diagnostic odyssey. However, in Pakistan, availability of the limited resources restrains the use of NGS for diagnostic purposes.

We have added this information in the discussion part.

Reviewer 3 Report

The authors presented a nice manuscript disclosing clinical and genetic aspects of Parkistani families presenting with spinocerebellar disorders. Although some of the cases do not show pure spinocerebellar ataxia phenotypes, the authors included cases in which their clinical suspicion was related to spastic ataxias, hereditary spastic paraplegias and ataxias with neuropathy. The manuscript is original and some points could be discussed and even improved by the authors: 

1. Did any of the patients with B4GALNT1 variants presented with dystonia, dyskinesias, scoliosis, pes cavus, or cataracts? As several of these signs have been observed previously in patients in the B4GALNT1-related CDG phenotype? 

2. Was neurophysiological testing with needle EMG and nerve conduction studies performed in the patients? Some of the cases presented with variants in genes typically associated with axonal polyneuropathy or lower motor neuron involvement and these cases would really benefit from these analysis which could also be a hallmark of the phenotype. 

3. Were the patients evaluated for fundoscopic changes? It would be important especially for individuals with sacsinopathies.

4. BICD2-related cases should be only described as SMALED type 2A if there is evidence of lower motor neuron disease.

5. All gene citations in the text should be presented in italics. Authors should revise grammar aspects in the manuscript (ie. changing "is" to "are" in the first sentence of the Abstract). 

Minor review of language aspects in the Abstract and in the manuscript's text. 

Author Response

Comments and Suggestions for Authors

The authors presented a nice manuscript disclosing clinical and genetic aspects of Parkistani families presenting with spinocerebellar disorders. Although some of the cases do not show pure spinocerebellar ataxia phenotypes, the authors included cases in which their clinical suspicion was related to spastic ataxias, hereditary spastic paraplegias and ataxias with neuropathy. The manuscript is original and some points could be discussed and even improved by the authors: 

  1. Did any of the patients with B4GALNT1variants presented with dystonia, dyskinesias, scoliosis, pes cavus, or cataracts? As several of these signs have been observed previously in patients in the B4GALNT1-related CDG phenotype? 

Response: No such symptoms are present, according to available information.

  1. Was neurophysiological testing with needle EMG and nerve conduction studies performed in the patients? Some of the cases presented with variants in genes typically associated with axonal polyneuropathy or lower motor neuron involvement and these cases would really benefit from these analysis which could also be a hallmark of the phenotype. 

Response: No neurophysiological tests were performed, due to limited resources availability.

  1. Were the patients evaluated for fundoscopic changes? It would be important especially for individuals with sacsinopathies.

Response: No, patients were not evaluated for fundoscopic changes

  1. BICD2-related cases should be only described as SMALED type 2A if there is evidence of lower motor neuron disease.

Response: Lower motor neuron disease symptoms were observed in BICD2 patients, especially in patients V:3 and V:4. Please have a look at Table 1, we have added this information.

  1. All gene citations in the text should be presented in italics. Authors should revise grammar aspects in the manuscript (ie. changing "is" to "are" in the first sentence of the Abstract). 

Response: corrected

Comments on the Quality of English Language

Minor review of language aspects in the Abstract and in the manuscript's text. 

Response: Thank you for your feedback on the quality of English language, manuscript was read by all co-authors and some of them are native English speakers, the quality of English language fine according to all co-authors.
